# MicroRNA characteristics in epithelial ovarian cancer

Kira Philipsen Prahm[1,2]*, Claus Kim Høgdall[2], Mona Aarenstrup Karlsen[1,2], Ib Jarle Christensen[1], Guy Wayne Novotny[1], Estrid Høgdall[1]

1 Department of Pathology, Molecular unit, Danish Cancer Biobank, Herlev University Hospital, Herlev, Denmark, 2 Department of Gynecology, Copenhagen University Hospital, Rigshospitalet, Copenhagen, Denmark

* kira@prahm.dk

**Data Availability Statement:** Data availability The full microarray data are available from the NCBI Expression Omnibus Database with the accession number: GSE94320. Clinical data are provided in Supporting Information files.

## Abstract

The purpose of the current study was to clarify differences in microRNA expression according to clinicopathological characteristics, and to investigate if miRNA profiles could predict cytoreductive outcome in patients with FIGO stage IIIC and IV ovarian cancer. Patients enrolled in the Pelvic Mass study between 2004 and 2010, diagnosed and surgically treated for epithelial ovarian cancer, were used for investigation. MicroRNA was profiled from tumour tissue with global microRNA microarray analysis. Differences in miRNA expression profiles were analysed according to histologic subtype, FIGO stage, tumour grade, type I or II tumours and result of primary cytoreductive surgery. One microRNA, miR-130a, which was found to be associated with serous histology and advanced FIGO stage, was also validated using data from external cohorts. Another seven microRNAs (miR-34a, miR-455-3p, miR-595, miR-1301, miR-146-5p, 193a-5p, miR-939) were found to be significantly associated with the clinicopathological characteristics ($p \leq 0.001$), in our data, but mere not similarly significant when tested against external cohorts. Further validation in comparable cohorts, with microRNA profiled using newest and similar methods are warranted.

## Introduction

Ovarian cancer (OC) is the fifth most common type of cancer death for women, and the most lethal gynaecologic cancer in the Western world [1]. The majority of patients (70%) are diagnosed in advanced stages (International Federation of Gynaecologic Oncology (FIGO) stage III-IV) having a 5-year survival rate less than 30% [2].

OC is known to be a heterogenic disease, consisting of four major histologic subtypes; serous, endometrioid, mucinous and clear cell tumours, with serous subtypes being the most common (approx. 70%) [2, 3]. Current data indicates that all of the histologic subtypes possess different morphologic and genetic alterations [4]. Previously most OCs were assumed to arise from cells lining the surface of the ovary; however, more recent studies suggest that many OCs may originate from cells of the fallopian tube [5, 6]. Additionally, with improved knowledge on morphologic and molecular genetic features of the subtypes, a two-tier grading system,

**Funding:** Funding the study was supported by: The Mermaid Project, awarded to CH and EH, https://www.mermaidprojektet.dk/en/frontpage/. Danish Cancer Research Foundation, awarded to KPP, https://www.dansk-kraeftforsknings-fond.dk/. Herlev Hospital Research Council, awarded to EH, https://www.herlevhospital.dk/forskning. The funders had no role in study design, data collection and analysis, decisions to publish, or preparation of the manuscript. Medical Prognosis Institute A/S (MPI), https://medical-prognosis.com/, performed the microarray analyses and provided study reagents for the analyses. MPI was not engaged in any other part of the scientific content of this paper, or in the decision to submit the manuscript for publication. Their participation does not alter our adherence to PLOS ONE policies on sharing data and materials.

**Competing interests:** The authors have declared that no competing interests exist.

dividing the OC patients in type I and type II tumours has been more and more accepted within the last decade [7–9]. Type I tumours are usually diagnosed in earlier stages, are less aggressive, and rarely harbour TP53 mutations. By contrast, type II tumours usually present in advanced stages, are more aggressive, unstable, often contain TP53 mutations and usually have a much worse prognosis [8, 9]. However, a recent, large, study have clarified that tumor grade is still of great importance, and high grade type I tumors shouldn't be mistaken as indolent tumors [10]. Nonetheless, a better understanding of both histologic subtypes and other bio-pathological features that characterize the cancer is still needed.

Standard treatment of epithelial OC (EOC) patients is primary cytoreductive surgery followed by adjuvant platinum-based chemotherapy [11]. However, as the patients' prognosis greatly depends on the cytoreductive result, with residual disease as the most important independent prognostic factor, the goal of primary treatment is to achieve complete cytoreductive surgery (no macroscopic visual tumour left) [12, 13]. For patients where cytoreductive surgery is considered impossible due to extensive disease, advanced age or a poor medical condition, neoadjuvant chemotherapy (NACT) followed by interval debulking surgery and adjuvant chemotherapy is a viable alternative [14]. However, despite several attempts to identify predictors for the cytoreductive outcome, controversy about their predictive ability remains, and still none have been implemented in a clinical setting [15–17]. Some researchers have addressed the possibility that the cytoreductive result is a matter of tumour biology, rather than the surgical effort [18, 19]. Identifying a test that could predict the cytoreductive result, would be of great value for treatment decisions, and could potentially decrease the risk of unsuccessful primary cytoreductive surgeries.

MicroRNAs (miRNAs) are a group of small, non-coding RNA molecules of 19–25 nucleotides, that within the last decade have shown important regulatory functions in cancer [20]. They exert their function at the post-transcriptional level, by complementary binding with targets on messenger-RNAs (mRNA) resulting in mRNA translational inhibition, degradation or gene activation [21]. A single miRNA can have multiple target genes, and numerous studies have indicated that miRNAs are involved in cancer development [20–25], and can act as either oncogenes or tumour suppressor genes, depending on their targets [26]. Some studies have described differences in miRNA signatures according to OC histologic subtypes, however, most studies have a small study cohort or have used qRT-PCR for miRNA profiling, limiting the number of miRNAs analysed [27–29].

In our previous work, the same cohort that provides the basis of this study, was used to identify miRNAs associated with survival and progression in EOC, several miRNAs were identified and two miRNAs, associated with time to progression, were validated in external cohorts [30]. This study continues our work to advance the knowledge of miRNA as a biomarker for clinical outcomes in OC. And here the aim was to elucidate if EOC would reflect differences in miRNA signatures according to histologic subtypes, FIGO stages, tumour grade, and the dualistic tumour classification. Furthermore, we aimed to clarify if miRNAs are associated with the results of primary cytoreductive surgery.

## Material and methods

### Patients

Patients were diagnosed and surgically treated for EOC between October 2004 and January 2010. All patients were part of the prospective ongoing cohort study; *Pelvic Mass*, initiated at Department of Gynecology, Rigshospitalet, Copenhagen, Denmark, and the cohort has been used and are described in further details previously [31, 32]. The miRNAs were profiled and have been used for investigation of their predictive performance for sensitivity to

chemotherapy and survival outcomes, in previous studies [30, 31]. Clinical information was retrieved from the Danish Gynecologic Cancer Database having a coverage rate of 97% according to the latest annual report [33–35]. Inclusion criteria for the current study were: a histologic verified diagnose of EOC and primary cytoreductive surgery. Exclusion criteria were: histology other than EOC, neoadjuvant chemotherapy, insufficient tumour tissue for analysis, a previous or an active cancer other than EOC. Investigated endpoints and characteristics of interest were FIGO stage (early stages (I-II) vs. advanced stages (III-IV)), tumour grade (low grade 1 vs. high grade 2–3) [36], histologic subtype (serous vs. others) and type I vs. type II tumours. All low-grade serous carcinomas, endometrioid carcinomas, clear cell neoplasms, and mucinous carcinomas were defined as type I tumours, and all high-grade serous carcinomas were defined as type II tumours (Table 1). Patients with FIGO stage IIIC and IV were furthermore used in the analysis of the residual disease after cytoreductive surgery (complete cytoreduction (no macroscopic visual tumour) vs. residual tumour) (Table 1).

All patients were followed until death of any cause, emigration or until January 17th, 2015, which ever came first.

## Tissue

Tumour tissue from the primary operation, was stored as formalin fixed and paraffin embedded (FFPE) tissue in the Danish CancerBiobank [37]. A pathologist, specialized in gynaecologic pathology, has revised histologic diagnoses for all tissue samples. From Haematoxylin and Eosin staining, all tissue samples showed to have a tumour percent above 50.

## miRNA microarray analyses

The miRNA microarray analyses was profiled for a previous research project [31] and performed using Affymetrix GeneChip miRNA Array, according to manufactures instruction [38, 39]. In short, total RNA was extracted from 20 μm thick FFPE tumour sections using the RecoverAll Total Nuclei Acid Isolation Kit for FFPE samples (Ambion, Inc 2130 Woodward St. Austin, TX9).

After RNA isolation the RNA concentration was measured for all samples using a nanodrop procedure (NanoDrop 1000 Spectrophotometer) to ensure a sufficient yield. If a sample had a low RNA concentration, the sample was concentrated using Eppendorf Concentrator 5301 (SpeedVac). The miRNAs were then labelled using FlashTag HSR Biotin RNA Labelling Kit (Genishere, PA) and run on Affymetrix GeneChip 1.0 miRNA microarrays.

## Validation cohorts

For validation of our results we retrieved three publicly available datasets (GSE25204, GSE73582 and GSE73581) from the NCBI Gene Expression Omnibus database [40]. The datasets contain clinical information and miRNA array expression profiles from patients with EOC and have been described in detail in previous reports where they were used for investigation and validation [41–43]. The GSE25204 and GSE73582 datasets were combined and treated as one dataset. An overview of the cohorts is presented in S1 Table, and further details are described in S1 File.

## Data processing and statistical analyses

The miRNA array data was background adjusted, normalized and log2 transformed in R (R Development Core Team, Vienna, Austria, http://www.R-project.org) using the justRMA function in the Bioconductor package (the Quantile-Quantile method) [44]. MiRNA array data for

**Table 1. Baseline characteristics of 197 patients with epithelial ovarian cancer.**

| | |
|---|---|
| Status | |
|   Alive | 64 (32.5%) |
|   Death | 133 (67.5%) |
| Median age in years (range) | 64 (31–89) |
| Median OS in months | 48 (95% CI: 40–56) |
| Histology | |
|   Serous adenocarcinoma | 162 (82%) |
|     Low-grade | 4 (2.5%) |
|     High-grade | 158 (97.5%) |
|   Mucinous adenocarcinoma | 11 (6%) |
|   Endometrioid adenocarcinoma | 15 (8%) |
|     Low-grade | 9 (60%) |
|     High-grade | 6 (40%) |
|   Clear Cell adenocarcinoma | 9 (5%) |
| FIGO stage | |
|   IA | 22(11.2%) |
|   IB | 2 (1.0%) |
|   IC | 7 (3.6%) |
|   IIA | 3 (1.5%) |
|   IIB | 6 (3.0%) |
|   IIC | 12 6.1%) |
|   IIIA | 3 (1.5%) |
|   IIIB | 10 (5.1%) |
|   IIIC | 106 (53.8%) |
|   IV | 26 (13.2%) |
| Histologic grade | |
|   1 | 20 (10%) |
|   2 | 102 (52%) |
|   3 | 74 (38%) |
|   Unknown | 1 (<1%) |
| Type I or II tumours | |
| I | 39 (19.8%) |
| II | 158 (80.2%) |
| Residual tumour after surgery | |
|   0 | 94 (48%) |
|   <1 cm | 32 (16%) |
|   >1 cm ≤ 2 cm | 22 (11%) |
|   >2 cm | 49 (25%) |

OS = overall survival, FIGO = International Federation of Gynecology and Obstetrics

the external cohorts were already normalized and log2 transformed. Due to the log2 transformation, a rise of one in OR represents a twofold change in the miRNA expression level.

The complete miRNA data files, used in the current study, are available from the NCBI's Gene Expression Omnibus database (https://www.ncbi.nlm.nih.gov/geo/) under the accession number GSE94320 [40].

Preliminary, univariate logistic regression modelling of the expression level of 847 human miRNAs were performed for all outcomes. Multivariate logistic regression analyses were then

performed with the identified miRNAs form the univariate analyses, adjusted for all significant miRNAs. As only one miRNA was identified with a significant association to residual disease, no adjustment for other miRNAs were performed in multivariate analysis here, but the result was verified by 10-fold cross validation (Table 2).

To reduce potential overfitting the multivariate results were verified by ten-fold cross validation.

Due to small number of patients in some subgroups, histologic subtype, FIGO stage, tumour grade and result of cytoreductive surgery were treated as binary outcomes, as defined previously. Receiver-operating characteristics (ROC) were performed to assess the predictive value of identified miRNAs from the multivariate analyses.

For validation of our results, significant miRNAs identified in the multivariate results (Table 2) were tested in the external datasets (GSE25204+GSE73582 and GSE73581).

Statistical significance was for the preliminary univariate logistic regression analyses defined by a $P$-value $\leq 0.001$. In subsequent analyses a p-value $< 0.05$ was accepted as statistically significant. The statistical analyses were performed using SAS statistical software packaged (version 9.4, Cary N.C. USA), R (v 3.1.0 R Development Core team, Vienna, Austria, http://www.R-project.org) (package RMS), and IBM SPSS statistical software version 19.

### Ethics statement

The Pelvic Mass study was approved by the Danish Ethical Committee (KF01-227/03 and KF01-143/04). Methods were carried out in accordance with relevant guidelines and regulations. All patients included in the study were informed both verbally and in writing, and written consent was given before enrolment in the Pelvic Mass study, including use of their tissue for the purposes of research.

**Table 2. Multivariate logistic regression analysis of miRNAs associated with clinicopathological characteristics.**

|  | OR | 95% CI | P-value | AUC |
|---|---|---|---|---|
| **Histology (serous vs. others)** |  |  |  |  |
| miR-130a | 4.72 | 2.38–9.35 | <0.0001 | 0.94 |
| miR-34a | 0.15 | 0.08–0.31 | <0.0001 |  |
| miR-455-3p | 0.14 | 0.05–0.41 | 0.0003 |  |
| **FIGO Stage (III+IV vs. I+II)** |  |  |  |  |
| miR-130a | 1.93 | 1.35–2.75 | 0.0003 | 0.75 |
| miR-595 | 6.90 | 2.52–18.90 | 0.0002 |  |
| **Grade (2+3 vs. 1)** |  |  |  |  |
| miR-1301 | 5.38 | 2.00–14.47 | 0.0008 | 0.96 |
| miR-146b-5p | 3.65 | 1.97–6.75 | <0.0001 |  |
| miR-34a | 0.10 | 0.03–0.37 | 0.0007 |  |
| **Type II vs. I tumours** |  |  |  |  |
| miR-34a | 0.09 | 0.04–0.24 | <0.0001 | 0.94 |
| miR-193a-5p | 7.81 | 3.13–19.46 | <0.0001 |  |
| miR-455-3p | 0.04 | 0.02–0.15 | <0.00001 |  |
| **Residual disease (radical vs. non-radical)** |  |  |  |  |
| miR-939[1] | 0.32 | 0.17–0.59 | 0.0003 | 0.72 |

OR = odds ratio, CI = confidence interval, AUC = area under the curve, FIGO = Federation of International Gynaecologic Oncology.

[1]Un adjusted in multivariate analysis as, miR-939 were the only miRNA of significance for residual disease in the preliminary univariate analysis.

## Results

### Clinical and pathological findings

From the *Pelvic Mass Study*, the first 246 consecutively included patients with EOC were identified. 49 patients were excluded due to insufficient tumour material for analysis (n = 24), NACT or palliative care (n = 15), carcinosarcoma (n = 5), other active cancer (n = 3), questionable diagnosis at pathologic revision (n = 2). A total of 197 patients with EOC were eligible for inclusion in the study. Baseline characteristics and clinical features of the patients are summarized in Table 1. Histological subtypes were represented as expected, with 162 (82.2%) serous carcinomas, 15 (7.6%) endometrioid carcinomas, 11 (5.6%) mucinous carcinomas and nine (4.6%) clear cell carcinomas. Patients diagnosed in early stages amounted 52 (26.4%) and 145 (73.6%) were diagnosed in advanced stages. Low-grade tumours were found in 20 (10.2%) patients and 177 (89.8%) were found to be high-grade. 39 (19.8%) patients were categorized with type I tumours and 158 (80.2%) with type II tumours. Complete cytoreductive surgery was obtained in 94 (47.7%) of the patients.

### miRNA and clinicopathologic characteristics

In the preliminary analyses of miRNAs associated with clinicopathologic characteristics 64 miRNAs were found to correlate with histologic subtype, 9 with FIGO stage, 21 with tumour grade, and 62 with type I or II OC ($p \leq 0.001$) (S2 Table). After adjustment for all significant miRNAs, three miRNAs (miR-130a, miR-34a, miR-455-3p) maintained a significant association with histologic subtype (odds ratio (OR) = 4.72, $p<0.0001$; OR = 0.15, $p<0.0001$ and OR = 0.14, $p = 0.0003$). Two miRNAs (miR-130a and miR-595) were found to be associated with advanced FIGO stage (OR = 1.93, $p = 0.0003$ and OR = 6.99, $p = 0.0002$). Three miRNAs (miR-1301, miR-146b-5p, miR-34a) maintained a significant association with tumour grade (OR = 5.38, $p\leq0.0008$; OR = 3.65 $p<0.0001$; OR = 0.10, $p\leq0.0007$). And three miRNAs (miR-34a, miR-193a-5p, miR-455-3p) maintained a significant association with type I or II tumours (OR = 0.09, $p<0.0001$; OR = 7.81, $p<0.0001$; OR = 0.05, $p<0.0001$) (Table 2). The combined performance of the identified miRNAs for the different characteristics, measured by the AUC of the ROC curve was 0.94 for histologic subtype, 0.75 for FIGO stage, 0.96 for tumour grade, and 0.94 for type I or type II tumours (Table 2 and Fig 1). Mean expression levels and fold changes of investigated outcomes are presented in Table 3.

In validation of the miRNAs associated with histologic subtypes, FIGO stage and tumour grade, miR-130a was the only miRNA that resisted validation. MiR-130a was significantly validated for histology in both the GSE25204+GSE73581 cohort (OR = 1.76, 95% CI = 1.31–2.37, $p = 0.0002$) (Table 4) and the GSE73582 cohort (OR = 1.53, 95% CI = 1.15–2.05, $p = 0.0039$) (Table 5). MiR-130a was also significantly validated for FIGO stage in the GSE25204+-GSE73581 cohort (OR = 1.25, 95% CI = 1.00–1.56, $p = 0.049$) (Table 4), but not in the GSE73582 cohort (Table 5). Furthermore, miR-34a was found with an opposite association for histologic subtype in the GSE25204+GSE73581 cohort (Table 4). Neither of the remaining miRNAs were significantly validated (Tables 4 and 5). MiR-595 and miR-146b-5p were not analysed in the microarray profiling from the GSE73781 cohort, and therefore not available for validation in this cohort.

### miRNAs and surgical outcome

A subgroup of 132 patients with FIGO stage IIIC (n = 106) or IV (n = 26) were used in evaluation of result after cytoreductive surgery. Of these, complete cytoreductive surgery were obtained in 40 (30.3%) patients, whereas 92 (69.6%) had residual tumour.

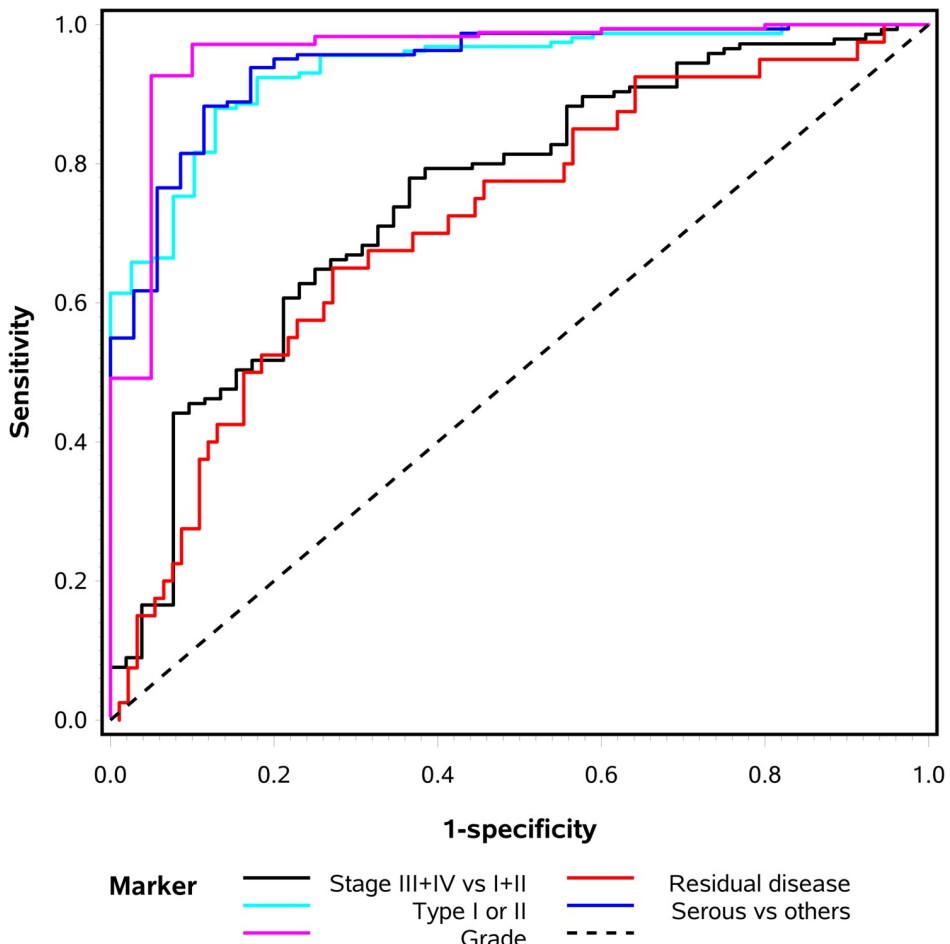

**Fig 1. Area under the curve (AUC) of receiver operating characteristic (ROC) of the combined miRNAs identified to be associated with FIGO stage, type I or II tumours, grade, residual disease and histology.**

In the preliminary univariate analysis, one miRNA (miR-939) was differentially expressed between patients who obtained complete cytoreductive surgery and patients who did not (OR = 0.32, $p$ = 0.0003) (S2 Table). The area under the curve (AUC) of the ROC for miR-939 corresponding to cytoreductive outcome was 0.72 (Table 2 and Fig 1). However, miR-939 did not remain significant in validation in the external cohorts (Tables 4 and 5).

## Discussion

In a prospective included cohort of Danish patients with EOC we investigated the association of miRNA profiling with clinicopathologic characteristics and result of cytoreductive surgery.

One miRNA, miR-130a, resisted validation, and was found to be significantly associated with a serous histologic subtype in both of the external cohorts, and with advanced FIGO stage in one of the external cohorts. Another seven miRNAs (miR-34a, miR-455-3p, miR-595, miR-1301, miR-146-5p, 193a-5p, miR-939) were identified as independently and significantly associated with clinicopathological characteristics and result of cytoreductive surgery in patients with EOC, but failed validation in external cohorts.

Substantial studies have shown that miRNAs are aberrantly expressed in cancer, including OC, and that they could act as either tumour suppressors or as oncogenes [45, 46]. Other studies

**Table 3. Mean normalised expression intensities and fold changes of miRNAs associated with histology, FIGO stage, grade and type I or II tumours.**

| | Mean expression intensity | | Fold change | 95% CI | P-value[1] |
|---|---|---|---|---|---|
| **Histology** | **Serous** | **others** | | | |
| miR-130a | 7.05 | 8.37 | 2.49 | 1.96–3.17 | <0.00001 |
| miR-34a | 8.00 | 7.28 | 0.61 | 0.47–0.66 | 0.00018 |
| miR-455-3p | 8.30 | 7.45 | 0.56 | 0.47–0.66 | <0.00001 |
| **FIGO stage** | **III-IV** | **I-II** | | | |
| miR-130a | 7.63 | 8.32 | 1.61 | 1.29–2.03 | 0.00005 |
| miR-595 | 1.13 | 1.39 | 1.20 | 1.10–1.31 | 0.00005 |
| **Grade** | **2+3** | **1** | | | |
| miR-1301 | 4.90 | 5.94 | 2.06 | 1.47–2.87 | 0.00003 |
| miR-146b-5p | 5.12 | 7.02 | 3.75 | 2.49–5.66 | <0.00001 |
| miR-34a | 8.41 | 7.30 | 0.46 | 0.33–0.64 | <0.00001 |
| **Tumour type** | **II** | **I** | | | |
| miR-34a | 8.05 | 7.25 | 0.57 | 0.45–0.73 | 0.00001 |
| miR-193a-5p | 6.57 | 7.25 | 1.60 | 1.32–1.94 | <0.00001 |
| miR-455-3p | 8.22 | 7.47 | 0.59 | 0.50–0.70 | 0.00009 |
| **Residual disease[2]** | **Radical** | **Non-radical** | | | |
| miR-939 | 5.27 | 4.75 | 0.70 | 0.58–0.83 | 0.00009 |

Expression intensities are on a log2 base, corresponding to a doubling of expression intensities per unit increase.

[1]P-values for the fold change are unadjusted (univariate).

[2]Values only evaluated in the 132 patients with FIGO stage IIIC and IV.

have previously demonstrated associations between miRNAs and clinicopathological features of OC, however, most studies were performed on smaller cohorts using qRT-PCR for the miRNA profiling [28, 29, 47], where only a few miRNAs are investigated, and few studies have used global miRNA microarray technology taking all known miRNAs in consideration [27, 48, 49].

MiR-130a which resisted validation in external cohorts and was found to be significantly associated with serous histology and advanced FIGO stage has also been described previously

**Table 4. Validation in the GSE25204+GSE73582 cohort (n = 263) of the miRNAs identified from multivariate analysis.**

| | OR | 95% CI | P-value |
|---|---|---|---|
| **Histology (serous vs. others)** | | | |
| miR-130a | 1.76 | 1.31–2.37 | 0.0002 |
| miR-34a | 1.13 | 0.82–1.55 | 0.46 |
| miR-455-3p | 0.46 | 0.38–0.55 | < .0001 |
| **FIGO stage (III+IV vs. I+II)** | | | |
| miR-130a | 1.25 | 1.00–1.56 | 0.049 |
| miR-595 | 0.85 | 0.56–1.30 | 0.46 |
| **Grade (2+3 vs. 1)** | | | |
| miR-1301 | 1.06 | 0.73–1.53 | 0.77 |
| miR-34a | 1.10 | 0.56–2.17 | 0.78 |
| miR-146b-5p | 0.83 | 0.45–1.52 | 0.55 |
| **Residual disease (radical vs. non-radical)** | | | |
| miR-939 | 1.05 | 0.68–1.61 | 0.83 |

OR = odds ratio. CI = confidence interval. FIGO = International Federation of Gynecological Oncology.

Table 5. Validation in the GSE73581 cohort (n = 179) of the miRNAs identified from multivariate analysis.

|  | OR | 95% CI | P-value |
|---|---|---|---|
| **Histology (serous vs. others)** | | | |
| miR-130a | 1.53 | 1.15–2.05 | 0.0039 |
| miR-34a | 0.84 | 0.60–1.18 | 0.32 |
| miR-455-3p | 0.91 | 0.67–1.25 | 0.57 |
| **FIGO stage (III+IV vs. I+II)** | | | |
| miR-130a | 1.18 | 0.93–1.48 | 0.17 |
| miR-595* | - | - | - |
| **Grade (2+3 vs. 1)** | | | |
| miR-1301 | 105.47 | 0.054 - >999.99 | 0.23 |
| miR-146b-5p[1] | - | - | - |
| miR-34a | 1.46 | 0.812–2.626 | 0.21 |
| **Residual disease (radical vs. non-radical)** | | | |
| miR-939 | 1.12 | 0.71–2.50 | 0.64 |

OR = odds ratio. CI = confidence interval. FIGO = International Federation of Gynecological Oncology
[1]Not available for analysis in validation cohorts

with aberrant expression in cancers [50, 51]. Several studies have demonstrated miR-130a to be associated with cisplatin resistance in OC, although with conflicting results, which could be due to different roles in different OC cell lines [52–54]. In vascular epithelial cell lines, miR-130a has shown to exert oncogenic functions, by interfering with genes that regulate angiogenesis [55]. The oncogenic features of miR-130a is in line with our results with higher expression in advanced FIGO stages.

MiR-34a, miR-130a and miR-455-3p that demonstrated significant associations with several outcomes in our primary analyses have all previously shown to exert oncogenic and tumour suppressing functions [51, 55–61]. In the present study miR-34a showed a significant association with histologic subtype, tumour grade and the dualistic tumour classification (type I or type II tumours). MiR-34a is directly regulated by the tumour-suppressor and transcription factor p53 and has shown to exert tumour-suppressing functions when it is introduced into cancer cells [56, 62, 63]. Consistent with previous studies we found miR-34a to be associated with non-serous histology, grade 1 and type I tumours, underlining the tumour-suppressive characteristics of miR-34a [64]. Abnormal expression of miR-455 has been reported for several cancers [58, 60, 65, 66], and one study found miR-455 to be downregulated in EOC tumour samples with low expression associated with advanced FIGO stages [59]. This corresponds to our findings, where miR-455-3p was lower expressed in the aggressive Type II tumours. The study was not able to demonstrate any association with histological subtypes; however, this was likely due to a small sample size (n = 45), with only 7 tumours of non-serous histology.

Complete cytoreductive surgery is an important prognostic factor for patients with advanced EOC [12, 67–69]. Similar survival rates have been observed for patients treated with primary cytoreductive surgery and chemotherapy compared to NACT and interval debulking surgery, in advanced FIGO stages [69–72]. However, observational studies have shown that patients who achieved complete cytoreductive surgery at primary surgery has improved survival compared to patients treated with NACT and interval debulking surgery [73, 74]. But today, no common uniform selection of patients for NACT exist. At multidisciplinary team conferences, usually a combination of the patients' performance status and evaluation of

ultrasound, computed tomography (CT) or positron emission-CT (PET-CT) are used in assessment of the optimal treatment choice. However, a recent review showed that CT has a poor predictive performance for residual disease [17]. FDG-PET/CT and MRI may have a better predictive performance, even so, only a limited number of studies have investigated the use on a routine basis [75–77]. Also, diagnostic laparoscopy has been suggested as a method for prediction of cytoreductive outcome. A recent randomized clinical trial, of 201 advanced stage OC randomized to either initial diagnostic laparoscopy or directly to primary cytoreductive surgery, demonstrated that initial diagnostic laparoscopy could reduce the number of futile primary cytoreductive surgeries with up to 29% [78]. However, the study has several limitations. First, radical surgery was defined as tumour < 1 cm, although the definition of complete cytoreductive surgery today is defined by no visible macroscopic tumour. Additionally, they included patient with suspected FIGO stage IIB or higher, although, complete cytoreductive surgery should be achievable in FIGO stage IIB-IIIB for surgeons with expertise in gynaecologic cancers. In contrast, a Cochrane review of laparoscopy for prediction of cytoreductive outcome, including seven studies with OC patients having FIGO stage IIB-IV, were unable to draw any firm conclusions, and included studies possessed similar limitation as the randomized trial [79].

In the current study miR-939 was found to be significantly associated with residual disease after primary surgery. Previously miR-939 has also been found as a promotor of cell proliferation in OC cells [80]. With a preoperative biopsy or maybe using ascites, miRNA profiling could be performed, and the expression level of miR-939 could potentially guide the choice of primary treatment. However, the same significant performance of miR-939 was not demonstrated in the validation cohorts, therefore our results must be confirmed in another study before any clinical use can be considered. Several other studies have investigated the tumour biology for surgical outcome with different genetic and molecular biomarkers. Studies of cancer antigen 125 (CA 125) have yielded conflicting results [81], while human epididymis protein 4 (HE4) have shown better performance but needs further validation [82, 83]. Also, gene expression has been explored as a possible predictor, but has mostly shown to be un-successful [18, 84]. Even so, one recent metanalysis including gene expression analysis from 13 studies, with a total of 1525 OC patient, where results were validated in independent samples, have shown more promising results. Here a gene signature was able to classify patients correctly in high- and low-risk of residual disease 92.8% of the time, with an AUC of 0.89 (95% CI = 0.84–0.93). However, this study also included all advanced stages of OC, and only optimal debulked (residual disease < 1cm) [85]. However, to our knowledge, no previous studies have investigated the association between miRNAs and surgical result in OC.

Our study has several strengths. First, the cohort consisted of a consecutive inclusion of patients, where all patients were operated and treated at the same tertiary centre by gynaecologic oncologists with high expertise. All clinical data were continuously updated, none of the patients were lost to follow-up, and missing data is limited. The representation of tumors according to histologic subtype, grade and FIGO stage matched the expected clinical representation of the tumors [34], and we have a mature study, where the shortest follow-up of a patient still alive is 61 months. Furthermore, the size of our cohort was similar to the validation cohorts and miRNA was profiled by global microarray, investigating a broad spectrum of miRNAs. Additionally, several of the miRNAs identified as significant in our preliminary and multivariate analyses are miRNAs previously described with oncogenic or tumour suppressing functions (S2 Table). All, except one, of the miRNAs from the well described Oncomir-1 family, also known as the miR-17-92 cluster, were identified with significant association to our outcomes [86]. Increasing expression of the miRNAs (miR-17, miR-18a, miR-20a, miR-19b, miR-92a), from the Oncomir-1 family, demonstrated significant associations with the more

aggressive type II tumours (S2 Table). This strengthens the validity of our data. However, an important limitation is, that the validation was performed in external cohorts where miRNA profiling had been analysed by different microarray platforms, complicating the comparison between the studies. This limitation may explain the lack of accordance between our results and results from validation. No other publicly available data with miRNA microarray profiling from OC patients, profiled on Affymetrix miRNA microarrays was found. It should also be mentioned that the miRNAs in the multivariate model were identified by choosing the best cross validated candidates, however, the association between miRNAs is substantial for some (21 combinations showed a correlation coefficient greater than 0.9 (S3 Table) suggesting that the chosen signature is not necessarily unique.

Other reasons for the unsuccessful validation could be addressed to possible batch effects, caused by laboratory differences, variations in analyses methods, reagent lots and personnel differences [87]. Additionally, our study and the used validation cohorts have significant differences in the study populations; patients with complete cytoreductive surgery ranged from 48% in our cohort to 29% and 41% in the validation cohorts. Also, tumour grade varied significantly, with only 38% having grade 3 in our cohort compared with 67% and 70% in the validation cohorts. Furthermore, our cohort comprised a higher number of patients with serous adenocarcinomas (82% in our cohort vs. 69% and 72% in the validation cohorts). Additionally, patients in our cohort were included between 2004–2010, whereas today more expertise gathered at tertiary centres has increased the use of NACT and more aggressive surgery, which has improved treatment results with a larger number of patients obtaining complete cytoreductive surgery today compared to the amount in our cohort. However, this could not explain the reason for variation in the number of patients who achieved complete cytoreductive surgery between the cohorts, as our cohort had the highest degree of complete debulked patients.

The results were not validated with qRT-PCR, which could be addressed as a limitation. However, validation in external cohorts must be considered as a more reliable form of validation limiting the risk of repeating false results. Further, miRNA microarray expression has shown to be highly concordant when re-analyzed with qRT-PCR [88, 89], with correlation coefficients measuring from r = 0.986 to 0.994, depending on the normalization method [90].

As previously mentioned, gene expression has also been investigated as promising biomarkers for OC. And in a recent large metanalysis, including 15 studies and a total of 3769 women with high-grade serous OC, gene expressions were used to develop a prognostic gene expression signature, that could predict patients at high- and low risk of achieving 5-year survival. A future interesting study would be to look at gene expressions signatures for the investigated clinicopathologic outcomes in the current study [91].

## Conclusions

In conclusion, miR-130a was found to be associated with serous histology and advanced FIGO stage in patients with EOC, and additionally validated in in two external cohorts. Another seven miRNAs were found to be significantly associated with clinicopathological characteristics and residual disease after cytoreductive surgery in patients with EOC, but only in our own cohort.

MiR-130a may be of prognostic importance for patients with EOC. Inconsistent results in external validation of the remaining miRNAs may be due to differences in study population characteristics, the use of different microarray platforms for miRNA profiling, differences in analyses methods or interlaboratory variation. Further validation of the results is warranted, preferably in a cohort of EOC patients where miRNA profiling has been performed on similar microarray platforms.

## Supporting information

**S1 File. Supplementary methods.**
(DOCX)

**S1 Table. Baseline characteristics of the validation cohorts.**
(DOCX)

**S2 Table. Univariate logistic regression analysis of miRNAs associated with clinicopathologic characteristics.**
(DOCX)

**S3 Table. Combinations of miRNAs demonstrating a correlation greater than 0.9.**
(DOCX)

**S1 Data.**
(SAV)

## Acknowledgments

We thank pathologist Lotte Nedergaard for her confirmation of all included patients' histologic diagnoses. We are grateful to the Danish CancerBiobank and the Danish Gynecologic Cancer Database for making tissue and data available for use in the present study. We thank Medical Prognosis Institute for providing study reagents, and for the conduct of the microarray analyses.

## Author Contributions

**Conceptualization:** Kira Philipsen Prahm, Claus Kim Høgdall, Mona Aarenstrup Karlsen, Ib Jarle Christensen, Guy Wayne Novotny, Estrid Høgdall.

**Data curation:** Kira Philipsen Prahm, Claus Kim Høgdall, Mona Aarenstrup Karlsen, Ib Jarle Christensen, Estrid Høgdall.

**Formal analysis:** Kira Philipsen Prahm, Claus Kim Høgdall, Mona Aarenstrup Karlsen, Ib Jarle Christensen, Guy Wayne Novotny, Estrid Høgdall.

**Funding acquisition:** Kira Philipsen Prahm, Claus Kim Høgdall, Estrid Høgdall.

**Investigation:** Kira Philipsen Prahm, Claus Kim Høgdall, Mona Aarenstrup Karlsen, Ib Jarle Christensen, Guy Wayne Novotny, Estrid Høgdall.

**Methodology:** Kira Philipsen Prahm, Claus Kim Høgdall, Mona Aarenstrup Karlsen, Ib Jarle Christensen, Guy Wayne Novotny, Estrid Høgdall.

**Project administration:** Kira Philipsen Prahm, Claus Kim Høgdall, Estrid Høgdall.

**Resources:** Estrid Høgdall.

**Software:** Claus Kim Høgdall, Ib Jarle Christensen, Estrid Høgdall.

**Supervision:** Claus Kim Høgdall, Mona Aarenstrup Karlsen, Ib Jarle Christensen, Guy Wayne Novotny, Estrid Høgdall.

**Validation:** Kira Philipsen Prahm, Claus Kim Høgdall, Ib Jarle Christensen, Estrid Høgdall.

**Visualization:** Claus Kim Høgdall, Mona Aarenstrup Karlsen, Ib Jarle Christensen, Estrid Høgdall.

**Writing – original draft:** Kira Philipsen Prahm.

**Writing – review & editing:** Claus Kim Høgdall, Mona Aarenstrup Karlsen, Ib Jarle Christensen, Guy Wayne Novotny, Estrid Høgdall.

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
