## [Decision Letter · Decision Letter 0]

27 Oct 2020

PONE-D-20-23009

MicroRNA characteristics in epithelial ovarian cancer.

PLOS ONE

Dear Dr. Prahm,

Thank you for submitting your manuscript to PLOS ONE. After careful consideration, we feel that it has merit but does not fully meet PLOS ONE’s publication criteria as it currently stands. Therefore, we invite you to submit a revised version of the manuscript that addresses the points raised during the review process.

In addition to addressing the comments from the reviewers, please focus on the validation of the experiments and means of accessing rigor and reproducibility.

We look forward to receiving your revised manuscript.

Kind regards,

Shannon M. Hawkins, M.D., Ph.D.

Academic Editor

PLOS ONE

Journal Requirements:

2. Please provide additional details regarding participant consent. In the ethics statement in the Methods and online submission information, please ensure that you have specified whether consent was informed regarding the use of their tissue for the purposes of research.

3. Please note that PLOS does not permit references to “data not shown.” Authors should provide the relevant data within the manuscript, the Supporting Information files, or in a public repository. If the data are not a core part of the research study being presented, we ask that authors remove any references to these data.

4. To comply with PLOS ONE submission guidelines, in your Methods section, please provide additional information regarding your statistical analyses. For more information on PLOS ONE's expectations for statistical reporting, please see https://journals.plos.org/plosone/s/submission-guidelines.#loc-statistical-reporting.

5. In the Introduction section, please discuss how your study relates to and represents a scientific advance from your previous study in PLOS ONE (reference provided below):

https://journals.plos.org/plosone/article?id=10.1371/journal.pone.0207319

Please provide a sound scientific rationale and justification for a separate publication on this topic. Thank you for your attention to this request.

Additional Editor Comments (if provided):

Reviewers' comments:

Reviewer's Responses to Questions

**Comments to the Author**

1. Is the manuscript technically sound, and do the data support the conclusions?

Reviewer #1: Partly

Reviewer #2: Yes

2. Has the statistical analysis been performed appropriately and rigorously? 

Reviewer #1: Yes

Reviewer #2: Yes

3. Have the authors made all data underlying the findings in their manuscript fully available?

Reviewer #1: Yes

Reviewer #2: Yes

4. Is the manuscript presented in an intelligible fashion and written in standard English?

Reviewer #1: Yes

Reviewer #2: Yes

5. Review Comments to the Author

Reviewer #1: The study performed by Prahm et al measured the expression levels of microRNAs in tumor samples from 197 patients with epithelial ovarian cancer using miRNA microarray. Data were normalized and analyzed for comparison between patients with different FIGO stages, tumor grades, histologic subtypes, and type I and type II tumors. The identified miRNAs with significantly altered expression levels between different groups of patients were further validated by analyzing external datasets where miRNA array expression profiles of ovarian cancer patients were described. The manuscript was well written and the data were presented in an intelligible fashion. However, the finding is not novel and the significance of the manuscript is compromised.

Questions and criticism:

1. Most identified miRNAs with altered expression cannot be validated using the external datasets. Although this has been extensively discussed, it still brings concerns about the quality of the data. The microarray data are the only data created by this manuscript. The authors need to describe more about data processing, particularly how the raw data were normalized because miRNA arrays are usually spotted in low density due to low numbers and low expression of miRNAs.

2. The authors have not done any experiments to validate their array data except using the external datasets. But the external datasets failed to validate the changes of most miRNAs. Moreover, the fold changes of identified miRNAs are quite low, from 0.46 to 3.75 folds. A validation experiment using qRT-PCR is then highly needed. Other than confirming the changes of those identified miRNAs, the regulation of mRNA targets could be an alternative to validate the array data.

3. High-quality miRNA microarray data come from high-quality RNA samples. The authors have not mentioned anything about the quality control of their RNA isolation, particularly when RNA was isolated from formalin fixed and paraffin embedded tissues. But formalin fixation modifies nucleic acids and challenges the isolation of high-quality RNA for genetic profiling.

Reviewer #2: I found this manuscript to be very well written and clear in it's presentation. There are a number of things that I will point out, and I feel that all are addressable to make this work acceptable for publication. The biggest issues are the validation failures, which in all honesty will make the very good work by the authors to have quite diminished interest to the readership. Can the data be approached using a multiple miRNA panel to address the question that they raise: to investigate if miRNA profiles could predict cytoreductive outcome in patients with FIGO stage IIIC and IV ovarian cancer? This could be a simple as constructing a polynomial predicting probability of cytoreductive outcomes based on ALL of the miRNAs described here that perform well and compare this with the performance of anatomical clinicopathological characteristics. The recent work by a large international group has taken this type of approach (DOI:https://doi.org/10.1016/j.annonc.2020.05.019) using a panel of select genes that they have identified. Using a combined miRNA panel may overcome the potential problems that could result from determinations by external cohorts outside the authors group.

Using designations of Type I vs II ovarian cancers does not involve independent comparisons (see DOI: 10.3390/diagnostics10020056) because stage in addition to grade is a major determinant of aggressiveness.

Let there be no mistaking my review: a very major issue is the validation failures.

Additional points

1. While the origination of ovca in the fallopian tube has considerable momentum, the establishment that MOST ovarian cancers originate here is not well-established (line 55)

2. Lines 59-60 need to be adjusted in light of DOI: 10.3390/diagnostics10020056

3. Line 98 "are" change to "is"

4. Line 239 "differential" change to "differentially"

5. Line 295 "has" change to "have"

6. Line 362: "cloud" change to "could"

7. Reference 1 is outdated and should be replaced by https://doi.org/10.3322/caac.21590

8. A consideration of DOI:https://doi.org/10.1016/j.annonc.2020.05.019 should be included and referenced.

9. In Figure 1 the material below the figure is unclear to me. This should be explicitly explained in a figure legend. In addition, the colors chosen for the curves should be more distinct. For example, the 3rd and 4th curves from the top both appear black to me.

This work could be morphed into an outstanding contribution deserving publication and I encourage the authors to do so.

Otherwise the validation failures considerably reduce the importance of what they are reporting.

6. PLOS authors have the option to publish the peer review history of their article (what does this mean?). If published, this will include your full peer review and any attached files.

Reviewer #1: No

Reviewer #2: No

---

## [Author Response · Author response to Decision Letter 0]

27 Apr 2021

We are pleased that you have given us the opportunity to submit a revised manuscript. We appreciate the helpful comments and suggestions from editor and reviewers. A detailed response to each comment is presented in the following. All italic text are changes that have been made to the manuscript. We believe that the revision has improved the manuscript and hope you will find the revision satisfactory. 

All page and line numbers will refer to the manuscript with track changes.

Comments from Editors: 

Response: 

All PLOS ONE’s style requirements have now been carefully review and adjusted where needed. 

2. Please provide additional details regarding participant consent. In the ethics statement in the Methods and online submission information, please ensure that you have specified whether consent was informed regarding the use of their tissue for the purposes of research.

Response: 

The current sentence has now been added to the Ethic statement (page 9, line 198-199): 

“All patients included in the study were informed both verbally and in writing, and written consent was given before enrolment in the Pelvic Mass study, including use of their tissue for the purposes of research.” 

3. Please note that PLOS does not permit references to “data not shown.” Authors should provide the relevant data within the manuscript, the Supporting Information files, or in a public repository. If the data are not a core part of the research study being presented, we ask that authors remove any references to these data.

Response:

The previous sentence with “data not shown” on page 19, line 433) have now been changed to “(S3 Table)” and the data have been included as Supporting information files (S3 Table).

4. To comply with PLOS ONE submission guidelines, in your Methods section, please provide additional information regarding your statistical analyses. For more information on PLOS ONE's expectations for statistical reporting, please see https://journals.plos.org/plosone/s/submission-guidelines.#loc-statistical-reporting.

Response: 

In the method section, it has now been specified more precisely what method was used for the data processing; method section page 8, line 165-167: 

The miRNA array data was background adjusted, normalized and log2 transformed in R (R Development Core Team, Vienna, Austria, http://www.R-project.org) using the justRMA function in the Bioconductor package (the Quantile-Quantile method) (44).

Furthermore, in methods, under “Data processing and statistical analyses” (page 8, line 172) a direct link has now been added to the NCBI’s Gene Expression Omnibus database (https://www.ncbi.nlm.nih.gov/geo/). 

5. In the Introduction section, please discuss how your study relates to and represents a scientific advance from your previous study in PLOS ONE (reference provided below):

https://journals.plos.org/plosone/article?id=10.1371/journal.pone.0207319

Please provide a sound scientific rationale and justification for a separate publication on this topic. Thank you for your attention to this request.

Response: 

Thank you for your attention to our previous work, this is a relevant question.

During the last decade microRNA (miRNA) has been widely explored as potential biomarkers for cancer.

In our previous work in PLOS ONE, published in 2018, we used the same cohort of patients with epithelial ovarian cancer, to identify miRNAs associated with prognosis and/or resistance to chemotherapy in ovarian cancer that potentially could be used for predictive matters of prognosis and chemotherapy resistance. 

In the current study the aim was to elucidate if epithelial ovarian cancer possessed differences in miRNA profiles according to known clinicopathologic differences such as histologic subtype, FIGO stages, tumor grade and tumor classification. And additionally, to clarify if the miRNA profiles were associated with the patients result of their cytoreductive surgery. 

Potentially all results could have been included in one publication, but initially we planned to look at the harder outcomes (survival and chemotherapy resistance). However, we wouldn’t expect that miRNAs predictive of survival and treatment outcome, and potentially useful in personalized medicine, necessarily would be the same miRNAs that could be used for diagnosis, differentiation and treatment strategy. We believed that it would be complex to include it all in one paper, diminishing the results of the analyses. 

In the introduction page 4, line 96-100 we have now referred to our previous work and explained the advance of the current study:

“In our previous work the same cohort, that provides the basis of this study, was used to identify miRNAs associated with survival and progression in EOC, several miRNAs were identified and two miRNAs, associated with time to progression, resisted validation in external cohorts (29). This study continues our work to advance the knowledge of miRNA as a biomarker for clinical outcomes in OC. And here the aim was to elucidate if EOC would reflect differences in miRNA signatures according to histologic subtypes, FIGO stages, tumour grade, and the dualistic tumour classification. Furthermore, we aimed to clarify if miRNAs are associated with the results of primary cytoreductive surgery.” 

Reviewers' comments: 

Reviewer #1: 

The study performed by Prahm et al measured the expression levels of microRNAs in tumor samples from 197 patients with epithelial ovarian cancer using miRNA microarray. Data were normalized and analyzed for comparison between patients with different FIGO stages, tumor grades, histologic subtypes, and type I and type II tumors. The identified miRNAs with significantly altered expression levels between different groups of patients were further validated by analyzing external datasets where miRNA array expression profiles of ovarian cancer patients were described. The manuscript was well written and the data were presented in an intelligible fashion. However, the finding is not novel and the significance of the manuscript is compromised.

Questions and criticism:

1. Most identified miRNAs with altered expression cannot be validated using the external datasets. Although this has been extensively discussed, it still brings concerns about the quality of the data. The microarray data are the only data created by this manuscript. The authors need to describe more about data processing, particularly how the raw data were normalized because miRNA arrays are usually spotted in low density due to low numbers and low expression of miRNAs.

Response:

Thank you for your comment, we agree that the data processing could be more extensively described, and we have now underlined what method in R that has been used for the normalization; method section page 8, line 167: 

The miRNA array data was background adjusted, normalized and log2 transformed in R (R Development Core Team, Vienna, Austria, http://www.R-project.org) using the justRMA function in the Bioconductor package (the Quantile-Quantile method) (44).

2. The authors have not done any experiments to validate their array data except using the external datasets. But the external datasets failed to validate the changes of most miRNAs. Moreover, the fold changes of identified miRNAs are quite low, from 0.46 to 3.75 folds. A validation experiment using qRT-PCR is then highly needed. Other than confirming the changes of those identified miRNAs, the regulation of mRNA targets could be an alternative to validate the array data.

Response:

It is of course important to validate data to contribute to the integrity and usefulness of the published data available to the scientific community. The manner of validation should however be open for discussion as some methods typically used add surprisingly little information, with following limited usefulness for interpreting the data in the context of the subject investigated. The often-used qPCR for validation is now so concordant with array data (with up to r=0.994) that it is more a technical duplicate than an independent validation technique if used on the same material. The literature of ovarian cancer has many papers on miRNAs, however the agreement upon which miRNAs are important is not excessive, and in some cases even discrepant, indicating there is some fundamental difficulties or differences in how samples are classified which confounds the data despite presence of controls and validation within each paper. The strongest validation is to test material from an independent cohort, however this is often not feasible in terms of the time it will take to gather a sufficient number of samples. 

We have attempted to validate our data against published data from external cohorts, which we consider a strong validation if successful, and do actually validate a microRNA which additionally has been described in other ovarian cancer papers as well. The remaining miRNAs do not pass the validation with the external cohort, however most have been described in cancer papers in general, and even in ovarian cancers in particular, which indicates they could be important as well. 

3. High-quality miRNA microarray data come from high-quality RNA samples. The authors have not mentioned anything about the quality control of their RNA isolation, particularly when RNA was isolated from formalin fixed and paraffin embedded tissues. But formalin fixation modifies nucleic acids and challenges the isolation of high-quality RNA for genetic profiling.

Response:

We agree that this would be relevant information for the reader, and therefore we have now included the following in the method section page 7, line 147-149:

“After RNA isolation the RNA concentration was measured for all samples using a nanodrop procedure (NanoDrop 1000 Spectrophotometer) to ensure a sufficient yield. If a sample had a low RNA concentration, the sample was concentrated using Eppendorf Concentrator 5301 (SpeedVac).”

Reviewer #2: 

I found this manuscript to be very well written and clear in it's presentation. There are a number of things that I will point out, and I feel that all are addressable to make this work acceptable for publication. The biggest issues are the validation failures, which in all honesty will make the very good work by the authors to have quite diminished interest to the readership. Can the data be approached using a multiple miRNA panel to address the question that they raise: to investigate if miRNA profiles could predict cytoreductive outcome in patients with FIGO stage IIIC and IV ovarian cancer? This could be a simple as constructing a polynomial predicting probability of cytoreductive outcomes based on ALL of the miRNAs described here that perform well and compare this with the performance of anatomical clinicopathological characteristics. The recent work by a large international group has taken this type of approach (DOI:https://doi.org/10.1016/j.annonc.2020.05.019) using a panel of select genes that they have identified. Using a combined miRNA panel may overcome the potential problems that could result from determinations by external cohorts outside the authors group.

Using designations of Type I vs II ovarian cancers does not involve independent comparisons (see DOI: 10.3390/diagnostics10020056) because stage in addition to grade is a major determinant of aggressiveness.

Response: 

With the criteria used (p>0.001) for identification of miRNAs associated with cytoreductive outcome, only one significant miRNA was identified. Therefore, it’s not possible to construct a polynomial predicting probability as suggested, although it would be relevant. 

Let there be no mistaking my review: a very major issue is the validation failures.

Additional points

1. While the origination of ovca in the fallopian tube has considerable momentum, the establishment that MOST ovarian cancers originate here is not well-established (line 55)

Response: 

We agree and have changed the sentence to the following (page 3, line: 57-58): 

“Previously most OCs were assumed to arise from cells lining the surface of the ovary; however, more recent studies suggest that many OCs may originate from cells of the fallopian tube.”

2. Lines 59-60 need to be adjusted in light of DOI: 10.3390/diagnostics10020056

Response: 

Thank you for drawing our attention to this new article. After reading it, we have rewritten the paragraph and referred to this article (page 3, line: 62-68):

“Type I tumours are usually diagnosed in earlier stages, are less aggressive, and rarely harbour TP53 mutations. By contrast, type II tumours usually present in advanced stages, are more aggressive, unstable, often contain TP53 mutations and usually have a much worse prognosis (8, 9). However, a recent, large, study have clarified that tumor grade is still of great importance, and high grade type I tumors shouldn’t be mistaken as indolent tumors (10). Nonetheless, a better understanding of both histologic subtypes and other bio-pathological features that characterize the cancer is still needed.”

3. Line 98 "are" change to "is"

Response: we have not changed the sentence page 3, line 96-97: “Furthermore, we aimed to clarify if miRNAs are associated with the results of primary cytoreductive surgery.” As miRNAs are plural. 

4. Line 239 "differential" change to "differentially"

Response: Thank you, we have now corrected this (page 15, line 322). 

5. Line 295 "has" change to "have"

Response: Thank you, we have now corrected this (page 17, line 378). 

6. Line 362: "cloud" change to "could"

Response: Thank you, we have now corrected this (page 20, line 448).

7. Reference 1 is outdated and should be replaced by https://doi.org/10.3322/caac.21590

Response: Thank you, reference 1 has now been updated. 

8. A consideration of DOI:https://doi.org/10.1016/j.annonc.2020.05.019 should be included and referenced.

Response: This is an important and interesting study, we have described the study in the discussion (page 20, line 456-461):

“As previously mentioned, gene expression has also been investigated as promising biomarkers for OC. And in a recent large metanalysis, including 15 studies and a total of 3769 women with high-grade serous OC, gene expressions were used to develop a prognostic gene expression signature, that could predict patients at high- and low risk of achieving 5-year survival. A future interesting study would be to look at gene expressions signatures for the investigated clinicopathologic outcomes in the current study (91).” 

9. In Figure 1 the material below the figure is unclear to me. This should be explicitly explained in a figure legend. In addition, the colors chosen for the curves should be more distinct. For example, the 3rd and 4th curves from the top both appear black to me.

Response: The figure legend has been improved, and a more detailed description of the figure is now included (page 11, line 238-240):

“Fig 1. Area under the curve (AUC) of receiver operating characteristic (ROC) of the combined miRNAs identified to be associated with FIGO stage, type I or II tumours, grade, residual disease and histology.” 

Furthermore, the curves in the figure have been made thicker so they more easily are distinguished from each other (Fig. 1). 

This work could be morphed into an outstanding contribution deserving publication and I encourage the authors to do so.

Otherwise the validation failures considerably reduce the importance of what they are reporting.

---

## [Decision Letter · Decision Letter 1]

17 May 2021

MicroRNA characteristics in epithelial ovarian cancer.

PONE-D-20-23009R1

Dear Dr. Prahm,

We’re pleased to inform you that your manuscript has been judged scientifically suitable for publication and will be formally accepted for publication once it meets all outstanding technical requirements.

Kind regards,

Shannon M. Hawkins, M.D., Ph.D.

Academic Editor

PLOS ONE

Additional Editor Comments (optional):

Reviewers' comments:

Reviewer's Responses to Questions

**Comments to the Author**

1. If the authors have adequately addressed your comments raised in a previous round of review and you feel that this manuscript is now acceptable for publication, you may indicate that here to bypass the “Comments to the Author” section, enter your conflict of interest statement in the “Confidential to Editor” section, and submit your "Accept" recommendation.

Reviewer #1: All comments have been addressed

Reviewer #2: All comments have been addressed

2. Is the manuscript technically sound, and do the data support the conclusions?

Reviewer #1: Yes

Reviewer #2: Yes

3. Has the statistical analysis been performed appropriately and rigorously? 

Reviewer #1: Yes

Reviewer #2: Yes

4. Have the authors made all data underlying the findings in their manuscript fully available?

Reviewer #1: Yes

Reviewer #2: Yes

5. Is the manuscript presented in an intelligible fashion and written in standard English?

Reviewer #1: Yes

Reviewer #2: Yes

6. Review Comments to the Author

Reviewer #1: (No Response)

Reviewer #2: I appreciate the authors' efforts to make adjustments to the manuscript that reflect the points that I raised in review. I feel that it is ready for publication, having satisfied my review.

7. PLOS authors have the option to publish the peer review history of their article (what does this mean?). If published, this will include your full peer review and any attached files.

Reviewer #1: No

Reviewer #2: No

---

## [Editor Report · Acceptance letter]

27 May 2021

PONE-D-20-23009R1 

MicroRNA characteristics in epithelial ovarian cancer. 

Dear Dr. Prahm:

I'm pleased to inform you that your manuscript has been deemed suitable for publication in PLOS ONE. Congratulations! Your manuscript is now with our production department. 

Kind regards, 

on behalf of

Dr. Shannon M. Hawkins 

Academic Editor

PLOS ONE